# Living-Neuron-Based Autogenerator

**DOI:** 10.3390/s23167016

**Published:** 2023-08-08

**Authors:** Svetlana A. Gerasimova, Anna Beltyukova, Anastasia Fedulina, Maria Matveeva, Albina V. Lebedeva, Alexander N. Pisarchik

**Affiliations:** 1Department of Control Theory and System Dynamics, Neurotechnology Department, National Research Lobachevsky State University of Nizhny Novgorod, 603022 Nizhny Novgorod, Russia; 2Center for Biomedical Technology, Universidad Politécnica de Madrid, Pozuelo de Alarcón, 28223 Madrid, Spain

**Keywords:** autogenerator, close-loop control system, hippocampal slice, hybrid neural circuit, neuron-like oscillator

## Abstract

We present a novel closed-loop system designed to integrate biological and artificial neurons of the oscillatory type into a unified circuit. The system comprises an electronic circuit based on the FitzHugh-Nagumo model, which provides stimulation to living neurons in acute hippocampal mouse brain slices. The local field potentials generated by the living neurons trigger a transition in the FitzHugh–Nagumo circuit from an excitable state to an oscillatory mode, and in turn, the spikes produced by the electronic circuit synchronize with the living-neuron spikes. The key advantage of this hybrid electrobiological autogenerator lies in its capability to control biological neuron signals, which holds significant promise for diverse neuromorphic applications.

## 1. Introduction

Recent progress in solid-state microelectronics, molecular biology, and neuroscience has led to the development of neuromorphic devices, integrating artificial and living biological systems for the real-time monitoring and control of neural activity. Breakthroughs in control theory and nonlinear dynamics have driven the evolution of mathematical theories and physical models of neurons as dynamical systems, from individual elements [1,2,3] to complex neural networks [4,5]. Synchronization processes have played a crucial role in encoding and decoding neural dynamics, revealing the importance of self-organization processes in neuronal dynamics [6,7,8,9]. Furthermore, it has been demonstrated over time that the foundation of neuronal dynamics lies in the self-organization process of complex systems [10,11].

Advancements in machine learning applied to neural networks have addressed various problems, from simple calculations to predicting extreme events [12,13,14,15,16]. Brain–computer interfaces have been developed to restore and regulate brain neural activity, utilizing both invasive and noninvasive approaches [17,18,19]. These interfaces incorporate sophisticated sensor arrays and are applicable to different brain regions, such as the visual cortex, motor cortex, hippocampus, and others [20]. Technological advancements in physiological signal sensing systems have reduced the size while maintaining sensitivity, with wireless systems simplifying the recording and stimulation of signals [21]. Notably, a wireless and battery-free platform has been developed, enabling chronic real-time observation over several weeks and supporting closed-loop operation [22].

Another noteworthy example of a wireless and battery-free system comprises electronic circuits, a flexible electrode array, and a catecholamine sensor. This system facilitates in vivo experiments, allowing for the simultaneous optogenetic stimulation and electrochemical recordings of catecholamine dynamics in freely behaving subjects [23]. These platforms have the potential to enhance our understanding of physiological processes and create new research opportunities. Additionally, the integration of neurophotonics with prosthetics allows for high-speed communication between the brain and artificial limbs, optimizing interfaces and providing real-time sensory information to amputees through direct brain interfacing [24,25]. These developments advance neurobotics and foster improved interactions between neural systems and artificial devices.

Significant efforts have been made to establish unidirectional coupling between silicon structures and neurons, allowing for direct neural cell stimulation [26]. Nanoelectronic devices based on carbon nanotubes and nanowires can detect biomolecular chemical processes and bioelectrical activity in neurons [27,28]. Graphene transistors offer the ability to register, stimulate, or suppress nerve pulses on transparent and flexible substrates [29,30]. Multielectrode arrays show promise for cultivating and growing individual neuronal processes, leading to potential breakthroughs in understanding and manipulating neural activity [31]. These cellular-level advances hold exciting prospects for integrating silicon-based technologies and neurons [32,33].

In our work, we are particularly interested in close-loop systems, because they hold great promise as a means to recover or replace the activity of specific brain regions. These systems consist of complex control devices that stimulate certain areas of the brain, often those that have been damaged, with real-time feedback for correcting and optimizing the stimulation based on the specific task at hand [34]. Such technologies play a crucial role in applied medical research and in developing methods to restore neural connections and maintain normal brain activity. The relevance of this direction is underscored by the challenges in treating neurodegenerative diseases using conventional drugs, which often require invasive interventions directly into damaged brain structures.

Currently, electrical brain stimulation is increasingly utilized in clinical research, particularly in cases where synapses have been damaged due to injury or disease. This method fosters increased connectivity among neurons and facilitates the recovery of synaptic plasticity, leading to improved neurological conditions in patients [35]. As a result, close-loop systems are emerging as valuable tools in the quest for effective treatments and interventions to address brain-related disorders.

The development of invasive technologies for neural interfaces with closed-loop circuits has led to significant advancements in restoring neural connections. These devices offer finer and more precise stimulation settings that can adapt to changes in the physiological state, which is crucial to the recovery of nervous tissue functions. Closed-loop deep brain stimulation (CL-DBS) systems have emerged as a promising strategy for neuromodulatory treatment, as they integrate real-time brain-based feedback to fine-tune the applied stimulation [36].

For instance, in the treatment of patients with advanced Parkinson’s disease, deep brain stimulation (DBS) has been traditionally performed using open-loop stimulation, where electrical impulses are delivered continuously and independently of any feedback. However, recent developments have led to closed-loop DBS for Parkinson’s disease treatment, where stimulation is performed based on recorded and analyzed neuronal activity in real time [37]. Researchers such as Rosin et al. [38] have achieved significant progress by utilizing high-frequency deep feedback stimulation for Parkinson’s disease treatment in a primate model.

Similarly, other research groups have explored closed-loop DBS strategies for various neurological conditions. For example, a closed-loop digital biostimulator was developed to prevent hypersynchronous seizure activity in a cortical population model of epilepsy [39]. In the context of Alzheimer’s disease, closed-loop DBS strategies have been studied to gain insights into memory dynamics and potentially treat memory deficits in preclinical models [40].

Studies using animal models have shown the potential of closed-loop neuroprosthetic microdevices to improve functional connectivity among different areas of the cerebral cortex and promote rapid recovery from traumatic brain injury [41]. Compared with open-loop deep brain stimulation, the deep feedback stimulation method employed in these studies offers several advantages in the treatment of neurodegenerative diseases [42]. Thus, closed-loop systems hold tremendous potential for advancing the treatment and management of various neurological disorders.

Extensive studies aimed at enhancing motor control using spinal cord stimulation have been conducted in rodents [43] and non-human primates [44]. Subsequently, clinical trials in humans utilizing brain–spine interfaces have shown promise in potentially reversing paralysis [45,46]. The development of closed-loop systems involves the integration of cell cultures grown on microelectrode arrays, as exemplified by the neuromorphic prosthesis proposed in [47], where the authors highlight the potential of in vitro bidirectional neuromorphic systems for reprogramming the brain, achieved by building an interface between a biological neural network (cell culture) and a spiking neural network. These advancements offer promising prospects for treating a wide range of motor system diseases and neurodegenerative brain conditions.

The use of neuron cultures proves highly advantageous for developing various methods to replace and restore neuronal activity, as it enables the recording of potentials from different points on the multielectrode arrays. However, it is essential to acknowledge that ongoing research in modeling physiologically plausible neural cultures reveals variations in behavior among neural cell cultures from different brain areas. Nonetheless, the continued exploration of these avenues presents exciting opportunities to advance our understanding and treatment of neural disorders.

In our recent work, we proposed an optimal fiber-optic system for the active stimulation of artificial neurons [48]. This system utilized a FitzHugh–Nagumo (FHN) generator and an opto-electric communication channel. Through extensive experimentation, we identified effective stimulation parameters that enable the interaction between artificial and biological cells without compromising the biological viability of mouse brain hippocampal slices for several hours.

During our investigations, we successfully obtained and analyzed typical responses, such as evoked population spikes from pyramidal neurons, somas, and excitatory postsynaptic potentials from dendrites of pyramidal neurons, all observed during electrical stimulation of Shaffer collaterals. Building upon these findings, we developed a more complex adaptive stimulation system to facilitate the interaction of an artificial mini-network comprising two FHN neurons and a memristive device with a biological neural network [49]. Within this system, the memristive device, a metal–oxide–metal thin-film structure, functioned as an adaptive element, creating an active synapse within the artificial mini-network.

It is worth noting that our system operated as an open-loop system, lacking feedback mechanisms. As such, our previous developments [28,48,49] fall into the category of directed stimulation systems, which are now commonly referred to as open-loop systems [47]. The incorporation of a memristor as part of neural networks offers potential benefits in terms of energy efficiency, and its nanoscale size holds significant promise for the future development of neuroprostheses and neurochips. These innovations bring us closer to unlocking new possibilities in the field of neural interfaces and brain–machine interfaces.

In this paper, we present a prototype of a hybrid closed system that exhibits real-time self-organization capabilities. Traditional current injection circuits employed in neuron stimulation systems come with various limitations. One notable limitation is the fixed current amplitude, which may not be optimal for all types of neurons or stimulation scenarios, potentially leading to suboptimal efficacy or neuronal damage.

Additionally, the inherent single-channel configuration of these circuits restricts the number of neurons that can be simultaneously stimulated, presenting challenges for applications requiring stimulation of large neural networks or for research aiming at simultaneous stimulation of multiple neurons. Moreover, the circuit’s susceptibility to noise and interference may result in inconsistent or inaccurate stimulation, undermining the reliability of the system. Lastly, the circuit’s inflexibility in adapting to novel stimulation paradigms or research inquiries, necessitating manual parameter adjustment, limits its versatility and usability. While the conventional current injection circuit is widely used in neuron stimulation systems, its limitations prompt us to explore alternative stimulation approaches to overcome these shortcomings.

We define our system as a combination of an artificial FHN neuron and biological neurons from mouse brain hippocampal slice cells. Biological systems possess fundamental dynamical properties as active systems, with certain biochemical substances acting as distributed energy sources that support the propagation of waves, such as excitation waves in nerve tissues and heart muscles. In our approach, we aim to merge fundamental dynamical and physiological properties within a single biological system, leveraging the advantages offered by both components. This novel hybrid closed system holds potential to advance the field of neural interfaces and stimulate further research into more effective and versatile stimulation techniques.

## 2. Materials and Methods

The hybrid closed-loop system is composed of several key components: an analog electronic FHN neuron, biological neurons from the mouse brain hippocampus, a sophisticated system for monitoring and recording the functional state of the biological subject, and an oscilloscope with continuous-signal-recording capabilities.

Figure 1a illustrates the block scheme of an artificial neuron based on the FHN model. This electronic neuron functions as a relaxation threshold oscillator with cubic nonlinearity, achieved with the incorporation of diodes [26,28,49]. The FHN model, a simplified representation of the physiological Hodgkin–Huxley system, effectively captures the primary dynamic features of a biological neuron. Specifically, the FHN circuit is capable of operating in both excitable and oscillatory modes and facilitates smooth transitions between these modes. In this study, we utilize both excitable and oscillatory modes. 

The block scheme represents the FHN generator, emulating the neuron membrane capacitance, where u_common_ denotes the membrane potential. By using a potentiometer, the oscillatory circuit can generate pulse trains lasting 25–30 ms with frequencies ranging from 20 to 150 Hz. The pulse frequency can be adjusted with an external power source. In the hybrid electrobiological device, specific output signal parameters are employed: the stimulating pulse amplitude varies between 1.5 and 4 V; the frequency ranges from 20 to 65 Hz; and the pulse train duration lasts 25 ms. The output pulse is captured and displayed on the oscilloscope. As we utilize a threshold generator with two outputs, ground and output signal, the output pulse remains unmodified and is amplified using a standard operational amplifier circuit. The only variation introduced is in the amplitude of the pulse signal, which is adjusted using the potentiometer.

The comprehensive experimental protocols, data, and results concerning the interaction of rat brain neurons in hippocampal slices with the FHN generator have been extensively documented in our previous publication [26]. For a detailed account of the procedures and outcomes, we refer the readers to our earlier paper.

The dynamics of the electronic circuit is described by the FHN equations written in dimensionless form [1,2]:(1)$$\begin{array}{l} \frac{du}{dt}=f(u)-v\\ \frac{dv}{dt}=\varepsilon (g(u)-v)-I \end{array}$$
where *u* is the voltage associated with the fast evolution of the neuron transmembrane potential; *ν* is the current related to the slow sodium gating variable, which mimics ionic current; *f*(*u*) = *u* − *u*^3^/3 is the cubic nonlinearity provided by diodes D1–D6 (Figure 1); and *I* is proportional to the battery voltage and inversely proportional to the resistance of the potentiometer. Parameter *I* is associated with the neuron depolarization level characterizing the excitation threshold and can be adjusted by changing the resistance. *g*(*u*) is the piecewise linear function defined as *g*(*u*) = *αu* if *u* < 0, and *g*(*u*) = *βu*, if *u* ≥ 0, where *α* and *β* are the parameters responsible for the shape and location of the potential *V*-nullcline above the excitation threshold. Parameter *ε* determines relative time scales for variables *u* and *v*. 

The FHN circuit dynamics is determined by nonlinear functions *f*(*u*) and *g*(*u*), as displayed in Figure 2a.

Figure 3 illustrates the coupling scheme between the FHN generator and the biological cell ensemble, depicted by large blue rectangles. The signal generated by the FHN circuit is directed through an operational amplifier, indicated by a triangle, and then transmitted to the biological cell ensemble. Subsequently, the neuron’s signal is fed back to the FHN circuit through a resistor with a nominal value of 1 kΩ, establishing a closed-loop autogenerator.

The initial conditions employed in both the numerical simulations and real experiments were identical. Both the artificial and biological neural generators operated in excitable mode. In the coupled system, the output signal from the FHN circuit was fed into the input of the biological “oscillator” (stimulating electrode). Simultaneously, the output signal from the recording electrode was transmitted to the input of the FHN generator through the load resistance, as illustrated in the block scheme depicted in Figure 3. This configuration allowed for the dynamic interaction and feedback between the FHN circuit and the biological neurons, facilitating a comprehensive analysis of the coupled system’s behavior and responses.

The coupled system is modeled by the following equations, where index 1 denotes the electronic circuit and 2 refers to the neural generator:(2)$$\begin{array}{l} \frac{du_{1}}{dt}=f(u_{1})-v_{1}+u_{2}k\\ \frac{dv_{1}}{dt}=\varepsilon (g(u_{1})-v_{1})-I_{1}\\ \frac{du_{2}}{dt}=f(u_{2})-v_{2}+u_{1}d\\ \frac{dv_{2}}{dt}=\varepsilon (g(u_{2})-v_{2})-I_{2} \end{array}$$
where *u*_1_ and *u*_2_ represent potentials generated by the electronic and living neurons, respectively. The scaling relation between these variables is defined by attenuation coefficients *k* and *d*, which were determined in our experiment, where we measured millivolt input from biological neurons to the FHN circuit. The following parameter values were used: *I*_1_ = 0.19, *I*_2_ = 0.18, *ε*_1_ = 0.18, *ε*_2_ = 0.19, *α* = 0.5, *β* = 2. Note that parameters *k* and *d* are varied. 

In our study, we emphasize the significant utilization of the concept of resistive interconnection between the artificial- and living-neuron generators. This approach allows us to quantitatively illustrate the dynamics of the proposed system and qualitatively explain the effects from the perspective of nonlinear dynamics. It is crucial to note that our model intentionally disregards the biological concept of neuron populations; rather, it comprehensively describes the evolution of the entire system.

In particular, we model living neurons as oscillators generating 10–20 mV spikes at frequencies closely resembling the FHN oscillation frequency. This simplification enables us to focus on the core dynamics and interactions between the artificial and biological components, providing valuable insights into the behavior of the coupled system. By adopting this approach, we aim to elucidate the essential mechanisms at play, shedding light on the intriguing dynamics arising from the interplay of artificial and living neural elements.

In both the experimental setup and numerical simulations, we utilize only two control parameters: the signal amplification coefficient, *d*, and the FHN generator threshold, *I*_1_. This simplification allows us to reduce the system to two interacting oscillators, each with specific parameters corresponding to the experimental conditions. By adopting this approach, we aim to gain insights into the system’s behavior and explore its dynamics in a more manageable and controlled manner, enabling a clearer understanding of the interactions between living and artificial neurons.

All experiments were carried out on the C57BL/6 mouse line of 2–3 months of age. The scheme of the experimental setup is presented in Figure 4. Local field potentials (LFPs) were recorded on surviving sections of the hippocampus of 400 μm thickness, prepared using a vibratome (MicromHM 650 V; Thermo Fischer Scientific). Slices were permanently placed in artificial cerebrospinal fluid (ACSF) with the following composition (mM): NaCl, 127; KCl, 1; KH_2_PO_4_, 1.2; NaHCO_3_, 26; D-Glucose, 10; CaCl, 2.4; MgCl_2_, 1.3 (pH 7.4–7.6; Osmolarity 295 ± 5 mOsm). The solution was constantly saturated with a gas mixture containing 95% O_2_ and 5% CO_2_. Optical and electrophysiological methods were combined to record the LFPs. For the morphological analysis, a Scientifica SliceScope (Scientifica Ltd.) microscope with aerial 4X lens was used. The LFPs were recorded using a glass microelectrode (Sutter Instruments Model P-97) from the Harvard Apparatus (Capillares 30-0057 GC150F-10, 1.5 OD × 0.86 IK × 100 Lmm; Quantity: 225). The electrode was filled with the ACSF solution. The pipette tip resistance was 4–8 MΩ. It is a standard characteristic of using electrodes for the patch-clamp method [50]. The electrode was installed in the holder of a HEKA EPC 10 USB PROBE 1 preamplifier head connected to a HEKA EPC 10 USB biosignal amplifier. A stainless-steel bipolar electrode was used for electrical stimulation of the dentate gyrus area in the hippocampus. This electrode was connected to the FHN circuit output. Data visualization and recording were performed using Patchmacter software. Data processing was carried out using MATLAB software. Gaussian noise in the useful signal was filtered. 

The position of the stimulation electric field on the recording electrode and the optimal distance for its placement are both intriguing topics in the study of the spread of electrical signals. Earlier research has explored the behavior of electrodes in various environments and cells, and Peter Fromherz and his colleagues [51,52] have developed a particularly convenient form of electrode that is now utilized for the extracellular recording of potentials on multielectrode arrays. There exist theoretical studies on the extracellular activity of brain neurons under current stimulation, but at present, there is no successful physical prototype of such a system, leaving many questions open for further investigation.

Nonetheless, certain rules and protocols have been established for recording field potentials of brain neurons [53]. Optimal electrode placement involves locating them at a specific distance from each other to minimize both response and artifact. A greater distance among electrodes can lead to a reduction in the accuracy of recorded signals, so it is important to find the optimal distance for electrode placement.

In this study, we use a mouse hippocampus section as a biological object, since it has a strictly ordered structure and division into regions, which allows us to effectively use it in the construction of neurohybrid systems. The preparation of surviving sections of the hippocampus is one of the progressive models for experiments in various fields of natural sciences, in particular, electrophysiology. The use of such experimental model began in the second half of the 20th century. Preparation of such samples includes brain removal from a living animal and its placement in a cold solution of artificial cerebrospinal fluid to slow down the metabolic activity of neurons. After preparing the slices, they are left to incubate in a solution at a physiological temperature of 36–37 °C to restore the activity of neural cells [54]. 

This biological model has significant advantages over other models in the study of neuron functions and synaptic activity. First, conducting experiments on such samples facilitates work with optical microscopes, and the electrical stimulation technique and neuronal activity registration. This makes it possible to clearly install the electrodes in the areas of our interest, which is an advantage compared with in vivo experiments. Second, the cytoarchitecture of most synaptic contacts is preserved in such samples; the neurons have exactly the same structure and location as in the living brain in vivo, which allows one to obtain more biologically plausible results than, for example, in cellular neuron–astrocyte cultures, where cells are randomly arranged and have random synaptic contacts. Third, the use of living acute brain slice preparation excludes harmful effects on an animal that is conscious or under anesthesia, since artificial brain stimulation is often non-physiological and can cause injury and discomfort even to an animal under anesthesia experiments in vivo. 

The use of the hippocampus in electrophysiological experiments also has a number of advantages. The hippocampus has a clear structure, morphologically and functionally distinct layers that are clearly visible under a microscope. For example, you can distinctly see the location of the dentate fascia and its granular neurons, as well as the areas of the CA1-CA4 hippocampus divided into layers of *alveus covers*, *stratum pyramidale*, *stratum radiatum*, *stratum lacunosum*, *stratum moleculare* [55]. There is a three-synaptic or perforant information pathway in the hippocampus. The entorhinal cortex is the main entrance to the hippocampus from the neocortex; a signal received from the associated zones of the neocortex reaches the hippocampus. Some of the connections from the entorhinal cortex go directly to the CA1 field, other connections go to the dentate fascia, whose axons of granular neurons (mossy fibers) stimulate the CA4 and CA3 dendrites. The axons of the CA3 and CA4 zones are called Shaffer collaterals, which form contacts with the apical dendrites of the CA1 region of the hippocampus, whose neurons send their axons to the subiculum, from where the signal is sent back to the entorhinal cortex. This hippocampal structure has partially unidirectional synaptic connections that allow for accurate stimulation in the initial zone of the perforate pathway and registration of the synaptic response of neurons at the end of this pathway. This is a great advantage in the development of a closed-loop circuit of a living and artificial neural network.

To combine the mouse brain slice and the electronic neuron generator into a closed circuit, the following steps were carried out: First, the neuron-like generator was in excitable mode, and the biological neurons were inactive. Second, the signal from the FHN circuit output was sent to a stimulus electrode and then sent to the mouse brain neurons. Third, the received signal from the recording electrode was fed to the HEKA EPC10 USB data acquisition system. Finally, the signal from the Scope output using a double BNC cable was sent to the oscilloscope and input of the neuron-like generator.

## 3. Results

### 3.1. Modeling

The model of two coupled oscillators given by Equation (2) simulates our experiment with the living-neuron-based autogenerator from the viewpoint of nonlinear dynamics. The time evolution of Equation (2) is illustrated in Figure 5. Initially, both systems are in excitable mode (Figure 5a). The blue curve shows the FHN signal, and the red curve shows the modified FHN signal associated with neuron cell activity. When the feedback is switched on, the closed-loop system acts as an autogenerator (Figure 5b,c). As seen from the time series in Figure 5b, the periodic oscillations generated by the FHN electronic circuit and living neurons are synchronous, displaying the same frequency. This synchronization serves as compelling evidence that our autogenerator is functioning effectively. 

Note that the classical FHN system in the excitable regime has already been studied [56,57,58]. Here, we use parameters that bring the model as close as possible to our experimental system. Modeling a system with such parameters is of considerable interest for the formation of given pulse sequences for neurohybrid system applications.

### 3.2. Experiment

#### 3.2.1. FHN Circuit Results

At the beginning of our experiments, the hippocampal neurons were in a rest state, while the neuron-like generator was in excitable mode. An interesting effect was observed when the neuronal ensemble from the hippocampal slice transmitted a sufficient external voltage to exceed the necessary threshold for the occurrence of oscillations in the neuron-like generator. The biological neurons under a sufficiently strong critical stimulus started to generate spiking activity and in turn affected the FHN generator to change the frequency of the generated spikes. 

Figure 6a shows typical waveforms of artificial- and real-neuron activity. The blue color indicates the signal reaching the stimulus electrode from the neuron-like generator while operating in closed-loop mode, which leads to the oscillatory activity of the neuron-like generator. The red color shows the signal coming from the recording electrode and indicating biological-neuron activation. As seen from Figure 4b, there is synchrony between the stimulus and the response.

Figure 7 shows how the frequency of the neuron-like generator changes when the signal amplitude is increased. In the experiment, the FHN circuit is in self-oscillatory mode (20 Hz pulse frequency, 25 ms duration, 2 V amplitude). After closing the circuit (neuron-like generator–mouse hippocampal neurons–neuron-like generator), the neuron-like generator amplitude is increased to 4 V by changing the potentiometer resistance. Since the circuit is closed, changing the signal amplitude of the neuron-like generator leads to an increase in local field potentials, which in turn affect the neuron-like generator frequency. The frequency of the neuron-like generator signal steadily grows as the signal amplitude is increased.

#### 3.2.2. Biological Neuron Responses

Let us now examine the signal obtained when a large population of neurons fires simultaneously, as depicted in Figure 8. To analyze this signal, we utilize two indicators that characterize the electrical activity of the interconnected neuron group.

The first indicator is the amplitude of the stimulation-evoked potential, measured from the signal’s onset to its peak, representing the maximum amplitude of the local field potential. A more informative characteristic of neuronal activity is the slope or rate of increase in the stimulation-evoked potential, which reflects the activation rate of nerve fibers. This rate is measured as the tangent of the slope angle of the signal. The starting point is determined when the signal slope becomes linear, and the end measurement is taken when the slope deviates from linearity.

In situations where there is no communication loop, a delay is observed in the neuron’s response to the stimulus [49]. However, when operating in self-oscillating mode, there should be no such delay, as the system should demonstrate prompt responses to stimuli.

## 4. Discussion

The central concept behind this work was to develop a closed-loop oscillatory circuit capable of stimulating the activity of biological neurons. To achieve this objective, we ingeniously combined real neurons with a well-known neural-like generator based on the FHN model, forming a cohesive system in which the neural cells entrain to the FHN frequency. Through meticulous modeling, we effectively simulated the signal transmission process to the neuronal cells, enabling us to observe the fascinating self-organizing ability of the biological object within the integrated system.

Despite this achievement, one limitation of our current system is that the FHN oscillator’s output signal, in its unmodified form, generates a bipolar signal that can disrupt neural cells. Consequently, the cell responds to both positive and negative stimuli, making it challenging to distinguish artifacts from actual stimuli. We are optimistic that this issue will be addressed in our forthcoming experiments, where we plan to incorporate a modified unipolar FHN generator, as previously proposed [49]. This modification should mitigate the adverse effects and enhance the accuracy of stimulus detection.

Another significant challenge arises when the stimulation frequencies exceed 20 Hz. At this point, distinguishing the stimulus from artifacts becomes exceedingly difficult. This situation is clearly illustrated with the visualization of biological signals in Figure 7 and Figure 8.

The living-neuron-based autogenerator presents an intriguing and multifaceted appeal, drawing interest both from a fundamental perspective in exploring nonlinear dynamics and from an applied perspective for its potential for enabling the automatic tuning of parameters for self-organizing systems. The integration of electrical and biological systems onto a single chip is a rapidly evolving area of research with the potential to address challenges in neuroprosthetics and advance brain–machine interfaces.

Looking ahead, we have plans to extend our experiments to include a comprehensive study on the interaction of spiking artificial and biological neural networks. To achieve this, we aim to implement an oscillatory network based on memristive devices, leveraging their adaptive properties and energy efficiency. Our previous work on an open-loop interaction involving a mini-network of oscillators connected by a memrister and mouse brain slices has shown promising results, revealing the potential for adaptive stimulation of neurons based on the network’s dynamical state [28,49]. We foresee that the combination of biological and oscillatory neural networks into a single dynamical system may open up new avenues for encoding information through the dynamical states of the oscillatory network.

To further enhance the performance and stability of our system in future experiments, we propose the incorporation of fiber-optic technology [26,48,59]. This enhancement would serve to reduce noise and signal loss while granting us the flexibility to fine-tune frequency and pulse duration. By doing so, we could achieve a higher level of precision and reliability in our studies.

Overall, our work showcases the promising potential of integrating artificial neural-like circuits with living neurons to create dynamical closed-loop systems. While we acknowledge the challenges that lie ahead, we are confident that continued advancements and optimizations will fuel further breakthroughs in this hybrid approach, paving the way for exciting applications in neuroscientific research and beyond. The live oscillator concept and the integration of neural networks hold the promise of revolutionizing brain–machine interfaces and deepening our understanding of complex neural dynamics.

## 5. Conclusions

In this paper, we have presented the simplest autogenerating hybrid neural system for the first time to the best of our knowledge. We have succeeded in integrating biological neurons into an oscillatory electronic circuit, thereby creating a hybrid neural self-oscillatory system. Biological neurons regulate the frequency of the artificial neural generator signal, and it, in turn, stimulates their activity. We have demonstrated the switching of the neural oscillator from excitable mode to oscillatory mode when the entire system was closed. The neuron-like signal generated by the FHN circuit was used to stimulate living neurons. As a result, the obtained local field potentials had a shape similar to the stimulating signal. This is the characteristic of synchronized neural activity, which amplifies and weakens responses depending on the given neuron of a similar signal. Thus, here, we have presented two elements of a closed-loop circuit, a neuron-like oscillator based on FHN electronic neurons and a living hippocampal network, which operated in a coordinated synchronous manner.

We envision significant advancements in the field of biocompatibility and its application in neurochips [60]. The inclusion of various materials, such as silicon, platinum, iridium, gold wires with polyetherimide insulation, glassy carbon rods with peptide coating, carbon nanotubes, polymer-based electrodes, and graphene transistors, has played a crucial role in enhancing the compatibility of these devices with bio-objects. Moreover, the emergence of research modeling has enabled a deeper understanding of biological responses to these materials, paving the way for the development of highly effective neurochips.

The ultimate goal of our research is to create a sophisticated neurochip capable of independently adjusting its parameters to optimize the functionality of the brain’s neural network. We believe that this technological advancement could have a profound impact on individuals affected by neurodegenerative diseases, including the elderly; those with genetic predispositions; survivors of traumatic brain injuries; and patients with conditions like Alzheimer’s disease, Parkinson’s disease, amyotrophic lateral sclerosis (ALS), ganglionitis, among others.

Notably, deep nerve stimulation has already demonstrated considerable success in treating various neurodegenerative diseases [61,62,63]. For a new vagus nerve stimulator, preliminary clinical trials are necessary to determine the device’s efficacy and safety in treating specific diseases, as well as to establish optimal stimulation parameters. In such neuroprosthetic systems, the inclusion of feedback mechanisms and automatic adjustment capabilities would be highly desirable, further enhancing their therapeutic potential. Preliminary testing may include animal studies, including toxicological studies. Advances in this direction could help solve the urgent problem of restoring lost brain functions based on the cellular and network levels. 

It is crucial to acknowledge that using advanced neurochips for the central nervous system, especially complex structures like the hippocampus, presents significant challenges. Nevertheless, we are optimistic that our ongoing research will address these obstacles and contribute significantly to the field of neuroprosthetics. The integration of cutting-edge biocompatible materials and advancements in research modeling brings us closer to the development of a groundbreaking neurochip. Our efforts hold promise for transforming the treatment of neurodegenerative diseases and contributing to the field of neuroprosthetics.

Automatic adjustment and correction of parameters in real time represent an actively researched area of development for methods of neuromodulation, such as deep brain stimulation (DBS) and vagus nerve stimulation (VNS). These neuromodulation methods use electrodes to stimulate specific areas of the nervous system, and parameter adjustment can help optimize treatment efficacy and reduce side effects. Some studies have already shown that automatic adjustment of stimulation parameters can improve treatment efficacy and reduce side effects. For example, a study conducted in 2018 on patients with Parkinson’s disease showed that the use of an automatic DBS parameter correction algorithm improved motor control and reduced side effects [64].

However, despite the potential of this technology, its development and implementation in clinical practice are still in their early stages, and further research and clinical trials are needed to evaluate its efficacy and safety. In addition, accurate and reliable algorithms that can adapt to changes in a patient’s condition and effectively respond to them are necessary for the successful implementation of automatic parameter adjustment. It is also important to consider individual patient differences and their needs when choosing optimal stimulation parameters. Overall, the automatic adjustment of neuromodulation parameters can become a powerful tool for improving treatment efficacy and the quality of life for patients with various neurological and psychiatric disorders. However, continued research and development in this area are necessary for its successful implementation.

## Figures and Tables

**Figure 1 sensors-23-07016-f001:**
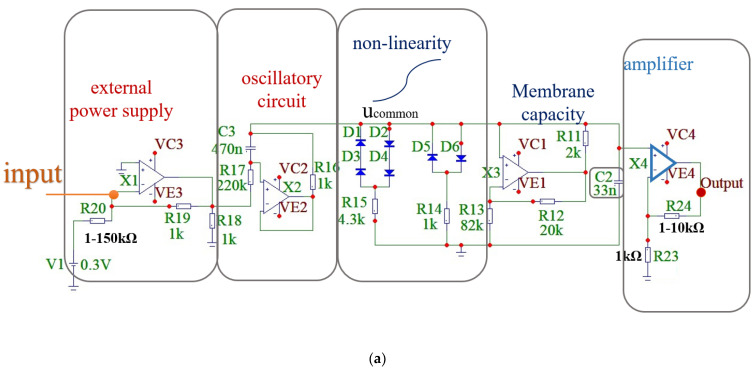
Electronic neuron and generated pulses. (**a**) Block diagram illustrating the FHN neuron-like generator, comprising an external source, electronic circuit, cubic nonlinearity element, and signal amplifier. Input and output points are denoted by red dots. (**b**) Representative waveform of the 30 Hz output signal.

**Figure 2 sensors-23-07016-f002:**
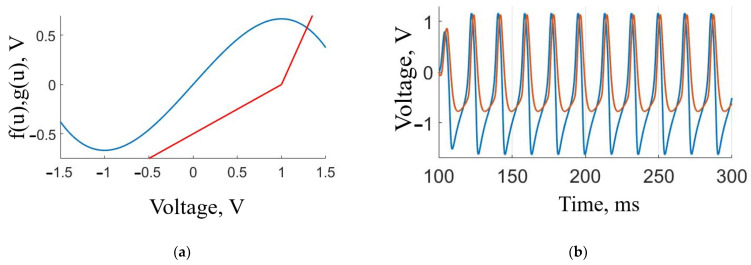
Characteristics of the FitzHugh–Nagumo model. (**a**) Smooth nonlinear *f*(*u*) (blue) and non-smooth double-linear *g*(*u*) (red) functions. (**b**) Time series of the 33 Hz pulses of *ν* (red) and *u* (blue) variables with 25 ms duration.

**Figure 3 sensors-23-07016-f003:**
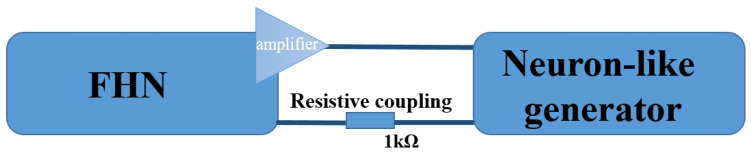
Illustrative block scheme of the concept of resistive mutual coupling between FiztHugh–Nagumo circuit (FHN) and biological “oscillator”.

**Figure 4 sensors-23-07016-f004:**
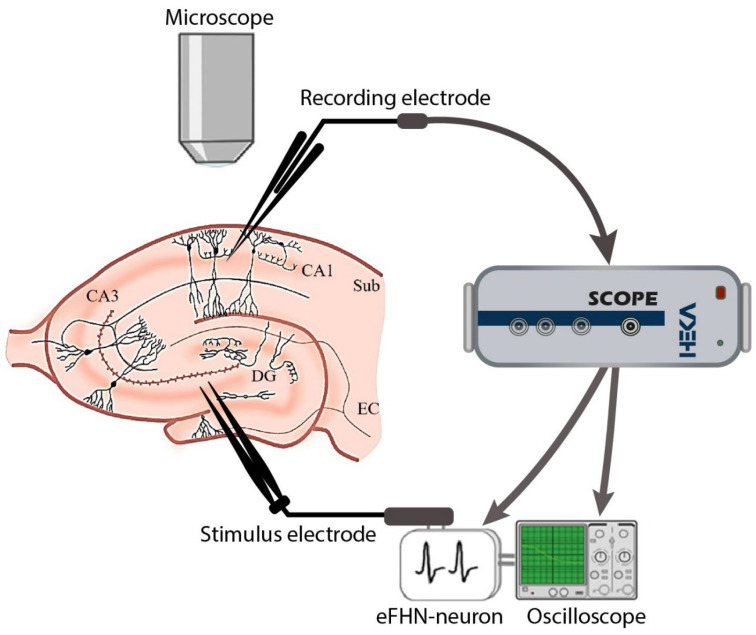
Experimental scheme with electrodes implanted in the hippocampal slice and pathway of signals from electronic neurons.

**Figure 5 sensors-23-07016-f005:**
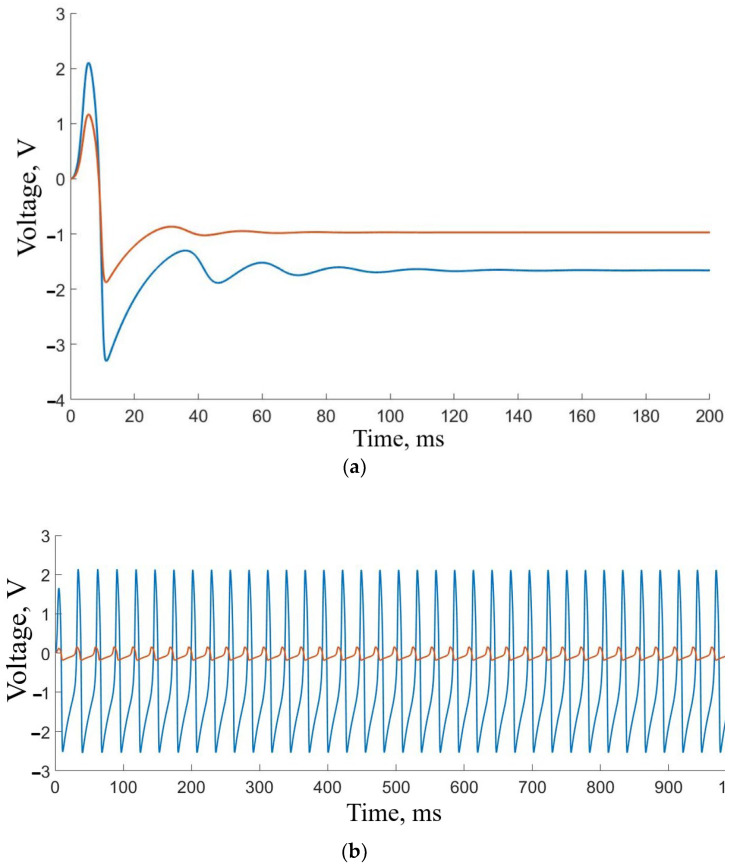
General illustration of the model neural activity. (**a**) Excitable regime corresponding to the initial system condition. (**b**) Time series of the neuron-like FHN signal (blue) and biological neuronal response (red) demonstrating a synchronous behavior. The “biological” signal is amplified by ten for better visualization. (**c**) Phase portrait of the voltages generated by the artificial and living neurons for d = 0.087 and k = 0.0223.

**Figure 6 sensors-23-07016-f006:**
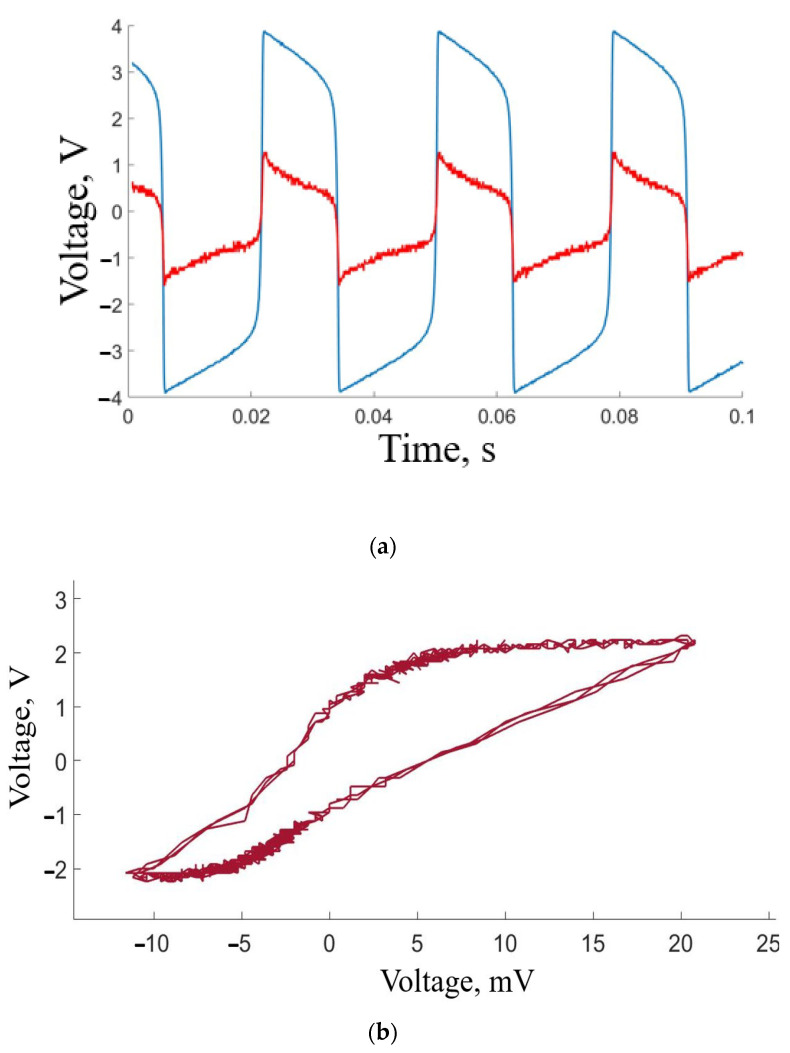
Illustration of neural activity. (**a**) Oscillograms demonstrating the signal from the neuron-like generator (blue) and the biological neuronal response (red), revealing synchronous spikes. (**b**) Phase portrait depicting the voltages generated by the electronic circuit and living neurons, providing a visual representation of their dynamical behavior and interaction.

**Figure 7 sensors-23-07016-f007:**
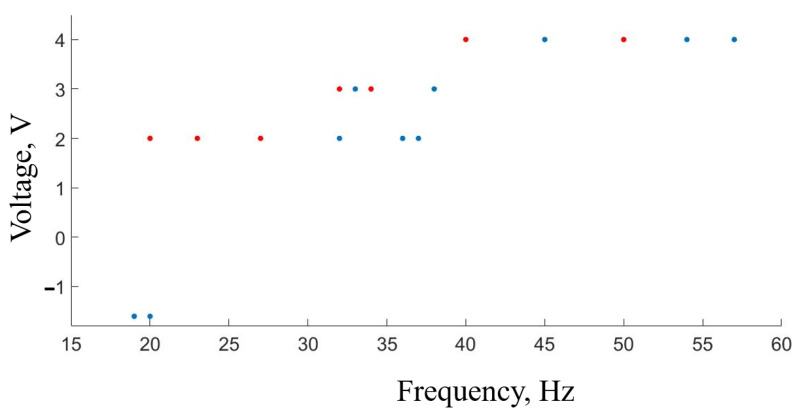
Changing the neuron-like generator frequency as the signal amplitude is increased. The red dots have the following initial conditions: oscillatory regime, pulse amplitude of 2 V, and a frequency of 20 Hz. The blue dots have the following initial conditions: excited regime with a voltage of −1.5 V.

**Figure 8 sensors-23-07016-f008:**
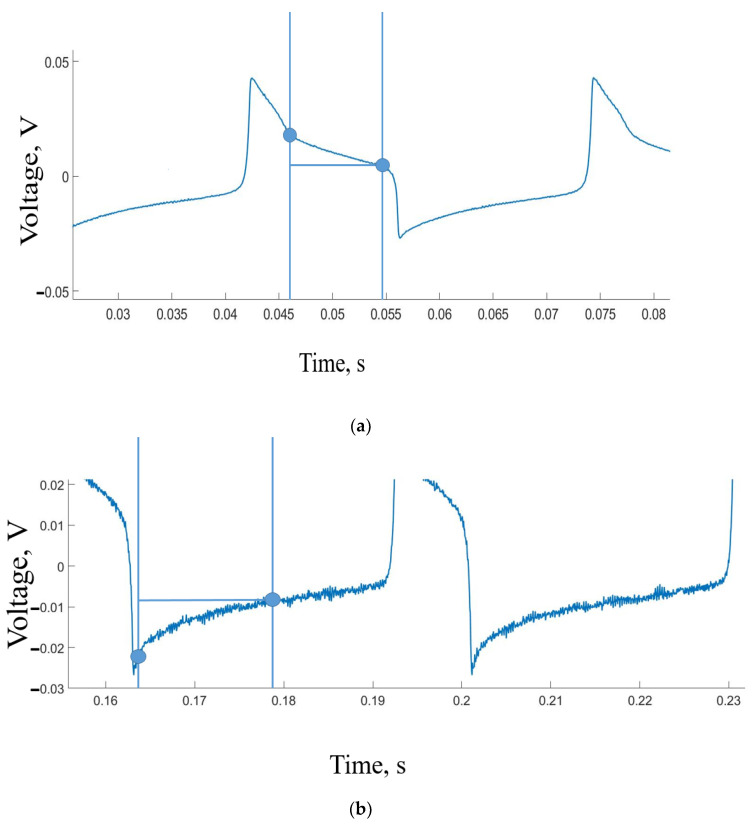
Examples of neuron responses. (**a**) Response on a positive peak, (**b**) response on a negative peak.

## Data Availability

The data are available at the following link: https://drive.google.com/drive/folders/1-mYVnls4WQVqujNWGGJR9hXItqBZZUUW (accessed on 30 November 2022).

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
