# Peer review of "Living-Neuron-Based Autogenerator"

_sensors, 2023, doi:10.3390/s23167016_

Round 1
Reviewer 1 Report (New Reviewer)
The authors present a hybrid system with the aim to control/modulate neural oscillations. Their results are investigated through simulation and mice experiments. The state of the art description is good. However, there are some things that require some clarification:
1. In Fig. 3, the coupling between the artificial and biological oscillators are illustrated. The figure is very important for the understanding of the results and discussion, but is not well discussed.
2. Another thing that could be improved is a discussion about how the simulation and the mice experiments differ from each other from the equation point of view. What are the variables that are the same? It is not clear.
3. In Fig. 5 (c), the authors say that it shows synchronization, but it is not clear. Could the authors clarify?
4. Another topic that requires clarification is the 'Biological neuron responses' section. I could not understand very well the discussion, specially the connection between the paragraph and the figure. Could the authors elaborate more?
Other Issues:
- The authors forgot to delete some sentences that are marked through the whole manuscript.
- Line 204, Fig. 1 is mentioned out of properly format.
- The voltage in the axis of the figures should be indicated in the figure, instead of the in the caption, for better understanding.
Despite of the comments, I believe the article is very interesting, it have many applications, and should be consider for publication after clarification of the mentioned topics.
Author Response
Comment:
The authors present a hybrid system with the aim to control/modulate neural oscillations. Their results are investigated through simulation and mice experiments. The state of the art description is good. However, there are some things that require some clarification:
ANSWER: We thank the Reviewer for all her/his valuable comments and suggestions. We revised the manuscript according to all Reviewer’s comments.
Comment 1: In Fig. 3, the coupling between the artificial and biological oscillators are illustrated. This figure is very important for the understanding of the results and discussion, but is not well discussed.
ANSWER: The discussion has been added.
Figure 3 shows how the coupling between the artificial and biological oscillators is realized. In the revised version, we have added the following discussion: “Figure 3 illustrates the coupling scheme between the FitzHugh-Nagumo generator (FHN) and the biological cell ensemble, depicted by large blue rectangles. The signal generated by the FHN circuit is directed through an operational amplifier, indicated as a triangle, and then transmitted to the biological cell ensemble. Subsequently, the neuron's signal is fed back to the FHN circuit through a resistor with a nominal value of 1 kΩ, establishing a closed-loop autogenerator.”
Comment 2: Another thing that could be improved is a discussion about how the simulation and the mice experiments differ from each other from the equation point of view. What are the variables that are the same? It is not clear.
ANSWER: The variables u1 and u2 in Eq. (2) represent potentials generated by the electronic and living neurons, respectively. The scaling relation between these variables is defined by the attenuation coefficients k and d, which were determined through our experiment, where we measured millivolts input from biological neurons to the FHN circuit. This corresponding comment has been added to the text.
Comment 3: In Fig. 5 (c), the authors say that it shows synchronization, but it is not clear. Could the authors clarify?
ANSWER: As evident from the time series in Fig. 5b, the periodic oscillations generated by the FHN circuit and living neurons are synchronous, displaying the same frequency. This synchronization serves as compelling evidence that our autogenerator is functioning effectively. This conclusion is supported by the time series shown in Fig. 5(b), rather than in Fig. 5(c). The corresponding comment has been added to the text.
Comment 4: Another topic that requires clarification is the 'Biological neuron responses' section. I could not understand very well the discussion, specially the connection between the paragraph and the figure. Could the authors elaborate more?
ANSWER: This section has been thoroughly elaborated. Furthermore, in Discussion we highlighted a notable limitation of our system. Specifically, when stimulated at frequencies higher than 20 Hz, distinguishing the stimulus from artifacts becomes exceedingly challenging. To visually demonstrate this situation, we have provided the visualization of biological signals in Figs. 7 and 8.
Comment: The authors forgot to delete some sentences that are marked through the whole manuscript.
ANSWER: Done.
Comment: Line 204, Fig. 1 is mentioned out of properly format.
ANSWER: Done.
Comment: The voltage in the axis of the figures should be indicated in the figure, instead of the in the caption, for better understanding.
ANSWER: Done.
Reviewer 2 Report (New Reviewer)
The introduction is a bit too lengthy, especially the very general first paragraphs.
For my opinion, the outlook in conclusion is a little too far-reaching, the authors should also consider the limitations of their study.
Minor flaws: Why are some sentences in main and figure text crossed out?
Only minor correction are needed by a native speaker.
Author Response
Comments: The introduction is a bit too lengthy, especially the very general first paragraphs.
ANSWER: We thank the Reviewer for his/her precious time, positive assessment of the manuscript and valuable comments, which has been addressed in the revised version. The first paragraphs of the introduction have been condensed.
Comments: For my opinion, the outlook in conclusion is a little too far-reaching, the authors should also consider the limitations of their study.
ANSWER: In the revised version, we highlighted the limitations of our study.
Comments:
Minor flaws: Why are some sentences in main and figure text crossed out?
ANSWER: Corrected.
This manuscript is a resubmission of an earlier submission. The following is a list of the peer review reports and author responses from that submission.
Round 1
Reviewer 1 Report
The authors developed a closed loop system for electrophysiology of brain slices, and realized the close loop autogenerator of mouse hippocampus through FHN model circuit. The content of the article tends to be a systematic attempt to put forward a conceptual prototype, which does not mean that the experiment proves whether the close loop will really generate autogenerator or the self-organization mentioned several times in the article. The major concern of this work is that it can not be proved that the influence of FHN model circuit on the hippocampus mainly originates from the artifact or autogenerator. The description in the article may be more inclined to the artifact. I think the author should first verify the reliability of the experimental results rather than publish them now. Some minor points are shown below.
1. There are no references in the fourth paragraph of the introduction.
2. Whether the 127 line in the penultimate paragraph of the introduction should be 'open loop'.
3. How does the impedance of brain tissue and internal power supply interact with this circuit. (Figure 1 shows how the output voltage, not current)
4. Figure 1 and Figure 3 should show the input and output curves.
5. Figure 4. Whether to exclude the effect of the stimulation electric field on the recording electrode and how far to put it (it seems to have an effect on the later results, and it may be an artifact as discussed).
6. Figure 5 (b) has no units, and why voltage is used to indicate no current.
7. Figure 6. Stimulation and neuron response are transient, and there is almost no delay. Why? (It may all be artifacts)
8. In Figure 8, can the impedance of 4-8M Ω measure such a good signal? Not very believable (it is possible that all the detected signals are input signals, and the burr inside is the real neural activity).
9. Under this condition, stable shock can also be achieved with open-loop stimulation. Is closed-loop necessary? (It seems unnecessary because they are all artifacts)
Reviewer 2 Report
Intro:
The authors devote a significant portion of their introduction to a review of notable current and past neural interface devices. The section covers functional electrical stimulation, DBS, etc., and other central and peripheral approaches in humans and preclinical models. It would be good if the authors could also include the latest in preclinical neuromodulation technologies such as the two following recent contributions for closed-loop preclinical studies in small animals:
doi.org/10.1021/acsnano.2c09475 and doi.org/10.1016/j.bios.2021.113886
Method and approach:
The design of the system, description of the approach and the biological experiments are satisfactory. The writing is clear and descriptive.
Discussion and conclusion:
The authors adequately address shortcoming and do indeed provide some insight into future works. There is also a section which speaks to recent progress in materials and technology which could facilitate future standalone device development.
The contribution of the manuscript would be improved is the authors could add more to the discussion and conclusion. Specifically it would be good to know what the different applications of this technology could be, and for what population. What form factor would the final device be. Finally, what preclinical work might be necessary prior to clinical applications.
Reviewer 3 Report
This paper presented an interesting work of FitzHugh-Nagumo model based neuron stimulation circuit and demonstrated “closed-loop” experiment on hippocampal. The relevant work will benefit both the physiological studies and other Neuromorphic applications. However, the present paper can be improved in the following aspects.
1).The introduction contains long background of the neuron stimulation with specific applications. But the limitations of the conventional current injection circuit based system have not been sufficiently discussed. This makes the motivation and exact novelties of the proposed system unclear. In addition, some of the statements can be improved to help understand the importance of the work. e.g. in line 133 “loses its original organization in the space and time”
2).In the method section. The mixed explanation of both the mathematical model and circuit makes it difficult to understand. I suggest the author introduce the mathematical model and the circuit separately. For example, what kind of circuit has been used to simulate the “nonlinear function”? Even this might be explained in the previous paper. It is necessary to briefly introduce it here as well.
Providing the validation of the mathematical model would benefit the overall statement. For example, the rationale of each equation in Eq 2 can be explained. Better to briefly explain how these equations are obtained and why it has been applied in the present study. Discuss the considerations behind the design will enhance the argument.
3).Figures can be improved. For example, Figures 1a and 3 are too abstract that makes people difficult to understand the structure of the system. I suggest to ether only use standard electronic symbol or just general blocks. In Fig 8, the waveform is clipped.
4).In the results, In Figure 6a, the overlapping of stimulation waveform and “neuronal response” makes me worried about the conclusion. I was expecting the lateral neuron response has some phase delay to indicate actual communications between the neurons.
I feel it is inappropriate to define the measured waveform as local field potential (Fig.8). Better to name it as “stimulation evoked potential”.